# Design of a Robust Flow Cytometric Approach for Phenotypical and Functional Analysis of Human Monocyte Subsets in Health and Disease

**DOI:** 10.3390/biom14101251

**Published:** 2024-10-03

**Authors:** Talia Ahrazoglu, Jennifer Isabel Kluczny, Patricia Kleimann, Lisa-Marie Irschfeld, Fabian Theodor Nienhaus, Florian Bönner, Norbert Gerdes, Sebastian Temme

**Affiliations:** 1Department of Anesthesiology, Faculty of Medicine, University Hospital, Heinrich-Heine-University, 40225 Düsseldorf, Germany; talia.ahrazoglu@hhu.de (T.A.); jennifer.kluczny@hhu.de (J.I.K.); 2Institute of Molecular Cardiology, Faculty of Medicine, University Hospital, Heinrich-Heine-University, 40225 Düsseldorf, Germany; patricia.kleimann@uni-duesseldorf.de; 3Department of Radiation Oncology, Faculty of Medicine, University Hospital, Heinrich-Heine-University, 40225 Düsseldorf, Germany; lisa-marie.irschfeld@med.uni-duesseldorf.de; 4Department of Cardiology, Pulmonology and Vascular Medicine, Faculty of Medicine, University Hospital, Heinrich-Heine University, 40225 Düsseldorf, Germany; fabiantheodor.nienhaus@med.uni-duesseldorf.de (F.T.N.); florian.boenner@krankenhaus-dueren.de (F.B.); gerdes@hhu.de (N.G.); 5Cardiovascular Research Institute Düsseldorf (CARID), Medical Faculty, Heinrich-Heine University, 40225 Düsseldorf, Germany

**Keywords:** monocyte subsets, flow cytometry, gating, myocardial infarction, coronary heart disease, phagocytosis, glucose uptake, MHC class II pathway

## Abstract

Human monocytes can be subdivided into phenotypically and functionally different classical, intermediate and non-classical monocytes according to the cell surface expression of CD14 and CD16. A precise identification and characterisation of monocyte subsets is necessary to unravel their role in inflammatory diseases. Here, we compared three different flow cytometric strategies (A–C) and found that strategy C, which included staining against CD11b, HLA-DR, CD14 and CD16, followed by several gating steps, most reliably identified monocyte subtypes in blood samples from healthy volunteers and from patients with stable coronary heart disease (CHD) or ST-elevation myocardial infarction (STEMI). Additionally, we established a fixation and permeabilisation protocol to enable the analysis of intracellular markers. We investigated the phagocytosis of lipid nanoparticles, the uptake of 2-NBD-glucose and the intracellular levels of CD74 and HLA-DM. This revealed that classical and intermediate monocytes from patients with STEMI showed the highest uptake of 2-NBD-glucose, whereas classical and intermediate monocytes from patients with CHD took up the largest amounts of lipid nanoparticles. Interestingly, intermediate monocytes had the highest expression level of HLA-DM. Taken together, we present a robust flow cytometric approach for the identification and functional characterisation of monocyte subtypes in healthy humans and patients with diseases.

## 1. Introduction

Monocytes, along with macrophages and dendritic cells, are part of the mononuclear phagocyte system (MPS) and play an important role in sterile injuries and in the control of infectious agents [1,2]. After formation in the bone marrow, monocytes circulate in the peripheral blood, where they comprise up to 10% of all peripheral leukocytes in humans. During inflammatory diseases, monocyte numbers in the blood strongly increase to initiate and maintain an appropriate immune response with the aim to efficiently counteract the injurious insult and to minimise organ damage [3].

Monocytes are a heterogeneous cell population and human monocytes within the blood can be classified into several different subsets [4]. For example, flow cytometric analysis of the cell surface markers CD14 (LPS co-receptor) and CD16 (low-affinity IgG receptor) [3] is used to differentiate three subtypes: (i) classical (CD14^++^/CD16^−^), intermediate (CD14^++^/CD16^+^) and non-classical monocytes (CD14^+^/CD16^++^) [3,5]. In humans, classical monocytes represent about 80–95% of circulating monocytes, while intermediate monocytes make up 2–8% and non-classical monocytes add up to 2–11% [6].

Deuterium labelling experiments in humans have revealed that monocyte subsets do not derive from distinct bone marrow progenitors but rather differentiate from classical monocytes after their release from the bone marrow (or the spleen) into the blood [7]. Classical monocytes reside in the blood stream for about one day until they differentiate into intermediate monocytes and circulate for approximately four days. Intermediate monocytes finally develop into non-classical monocytes that circulate in the blood for seven days. Inflammatory conditions stimulate haematopoiesis in the bone marrow but also trigger a rapid release of classical monocytes from the bone marrow into the blood stream. It has been shown that bone marrow-derived classical monocytes are the first cells that repopulate the circulation after an inflammatory event, whereas intermediate and non-classical monocytes appear at later time points [7].

In the case of an inflammatory lesion, monocytes migrate into the affected tissues, where they execute multiple effector functions such as phagocytosis of infectious pathogens or cellular debris, secretion of cytokines, chemokines and growth factors [8]. Mononuclear phagocytosis plays an important role during pathogen clearance and tissue remodelling by removing cell debris and extracellular matrix [9,10]. Additionally, monocytes extensively interact with other cell types of both the innate and the adaptive immune system [6]. After migration into the tissue, monocytes either undergo apoptosis or differentiation into tissue-specific macrophages or dendritic cells [11]. The variability of these effector functions within the inflamed tissue also depends on the particular monocyte subset [4]. Classical monocytes are among the first cells that are recruited from the bone marrow into the circulation and then into inflamed tissue, for example, via MCP-1 that is released by endothelial cells and fibroblasts [12]. In contrast, non-classical monocytes have been described to crawl on the endothelium to patrol the inner vessel wall in search of injuries [13,14]. It has been reported that non-classical monocytes are recruited either very early into the inflamed tissue, as for example in murine listeriosis infection, or in a second wave after classical monocytes [15]. Furthermore, monocyte subsets also seem to differently affect the migration of other cells. Chimen et al. found that, in a co-culture of non-classical and intermediate monocytes with neutrophils and endothelial cells, cell adhesion of neutrophils at the endothelium was promoted through TNF-α secretion. However, in a coculture of classical monocytes with endothelial cells, high levels of IL-6 were found, but blocking of IL-6 reduced neutrophil endothelial adhesion [16].

Classical monocytes are highly phagocytic, while intermediate monocytes preferably produce reactive oxygen species (ROS) and take part in antigen presentation and T cell activation [6]. All monocyte subsets secrete proinflammatory cytokines albeit with a different pattern. A study by Boyette et al. demonstrated the distinct secretion of IL-1β, IL-6 and TNF-α in monocyte subsets after overnight stimulation with toll-like receptor agonists [17].

Furthermore, particularly intermediate and non-classical monocytes seem to be important for antigen presentation and T cell activation [6]. This is also in agreement with the observation that non-classical monocytes and intermediate monocytes express the highest level of MHC class II molecules [3,5].

Since monocytes and their subsets play a pivotal role in multiple diseases, proper analysis of the phenotype and functionality of these cells can help to gain deeper insights into the underlying pathophysiological mechanisms. The most common way for the identification of monocyte subsets is flow cytometry, where monocytes are identified by size and granularity, which are represented by their forward/side scatter (FSC/SSC) properties, and/or CD11b, HLA-DR or CD86 expression [18,19,20,21]. Alternatively, extensive multi-marker and colour flow cytometry panels are employed, as described by Hally et al., using a 21-marker 18-color panel to identify monocyte subsets [22]. However, particularly under inflammatory conditions, the cell composition of the blood, the FSC/SSC properties and the expression levels of cell surface markers are altered, which can make the identification and analysis of monocyte subtypes challenging and result in contamination with non-monocytes like NK cells or B cells [23].

In the present study, we aimed to establish a robust protocol for the unequivocal identification of human monocyte subpopulations and to utilise this protocol for the analysis of functional aspects such as phagocytosis, glucose metabolism and the MHC class II pathway of antigen presentation in blood samples from healthy volunteers and patients with stable coronary heart disease (CHD) and ST-elevation myocardial infarction (STEMI).

## 2. Materials and Methods

### 2.1. Collection of Blood Samples and Study Population

Blood samples were collected from patients during hospitalisation in the department of cardiology at the University Hospital of Düsseldorf. Eligible criteria were informed consent and the presence of either stable coronary heart disease (CHD) or re-perfused ST-elevation myocardial infarction (STEMI). Stable CHD was defined as patients with a history of angiographically proven CHD who presented in the clinic for a routine checkup without any evidence for CHD progression. STEMI was defined as patients presenting with acute chest pain and evidence of ST-segment elevation in at least two contiguous leads and who were treated immediately with primary PCI (percutaneous coronary intervention). Blood samples from patients with STEMI were obtained on day one after myocardial infarction. This study conformed to the Declaration of Helsinki and was approved by the University of Düsseldorf Ethics Committee (File numbers: 2020-989, 2020-1052, 5961R [24]). Written informed consent was obtained from all patients. In total, samples from 23 healthy individuals (8 × male, 15 × female, age range 22–45, age median 24), 24 patients with STEMI (21 × male, 3 × female, age range 43–80, age median 59) and 10 patients with CHD (5 × male, 5 × female, age range 52–86, age median 64) were analysed.

### 2.2. Preparation and Characterisation of Fluorescent Lipid Nanoparticle

Fluorescently labelled perfluorocarbon nanoemulsions (^A488^PFCs) were prepared in house, as previously described [25,26]. Briefly, 0.24 g of the lipid mixture E80S (≙35 mM; Lipoid, Duisburg, Germany) was dissolved in 7 mL of phosphate glycerol buffer under constant motion for 30 min at room temperature. Subsequently, Atto488-labelled lipids (Atto488-DPPE, DPPE = 1,2-Dipalmitoyl-sn-glycero-3-phosphoethanolamin, Atto-Tec, Siegen, Germany) were added (100 µg per 10 g of PFCs) and stirred for a further 30 min at room temperature. After dissolving the lipids, 2 g of perfluoro-15-crown-5-ether (fluorochem, Hadfield, UK) (≙20% *w*/*v*) was added. Next, the weight of the suspension was adjusted to 10 g with phosphate buffer and, subsequently, a pre-emulsion was prepared using high-shear mixing (Ultra Turrax T18, IKA Werke, Staufen, Germany) for 3 min. The pre-emulsion was then further processed by microfluidisation (Microfluidizer LV1, Microfluidics, Newton, MA, USA) for five cycles at 1000 bar. The ^A488^PFCs were stored short term at 4 °C in the dark, or frozen at −20 °C for longer periods of time.

The characterisation of PFCs was performed using dynamic light scattering (Nanotrac Wave II, Microtrac Retsch, Haan, Germany). PFCs were diluted 1:10 in water (500 µL total volume) and measured three times with ten individual runs per analysis. Mean values of the three DLS measurements were calculated. The PFCs had a hydrodynamic diameter of 184.7 ± 3 nm (Appendix A). The polydispersity index (PDI) as an indicator for size distribution was 0.21 (±0.05), while the ζ-potential was −36.5 mV (±1.6 mV) (see Appendix A for graphical display of the size distribution).

### 2.3. Flow Cytometry

Flow cytometry was performed using a BD FACS Verse^TM^ and the corresponding BD FACSuite^TM^ software version V1.0.6 (Becton Dickinson, Heidelberg, Germany). The instrument was located in a climate and humidity-controlled laboratory and maintained regularly (cleaning routines, laser baseline settings). To verify consistency of the laser power and settings, CS&T^TM^ beads (BD Biosciences, Heidelberg, Germany) were run every morning prior to the first analysis. To prevent signal spill-over, individual compensation settings were performed for each fluorochrome and marker combination. Compensation was conducted either with compensation beads or single-labelled cells. To verify that the cells were analysed under constant flow, data-acquisition was started 5–10 s after the beginning of the flow cytometric measurements. The acquisition was run normally for 30–120 s and was always stopped before the tube was empty to avoid air bubbles entering the flow cell. Analysis of the flow cytometric data and preparation of the dot plots and histograms were performed using FlowJo V10.8.1 (FlowJo, Ashland, OR, USA) or Kaluza Analysis 2.2.1 (Beckman Coulter, Indianapolis, IN, USA). The gate boundaries were set using fluorescence-minus-one (FMO) controls for each marker of interest.

#### 2.3.1. Sample Preparation and Staining

An amount of 6 ml of peripheral venous blood was drawn into an EDTA tube (Becton Dickinson, Heidelberg, Germany) after sterile punction of the median cubital vein, using a 21 G butterfly (Becton Dickinson). Subsequently, blood was immediately mixed with 60 mL of erythrocyte lysis buffer (155 mM NH_4_Cl, 12 mM NaHCO_3_, 0.1 mM EDTA) and incubated for 10 min at room temperature.

Isolated cells were washed with 1 mL MACS-buffer [phosphate buffered saline (Sigma-Aldrich, St. Louis, MO, USA), 2 mM EDTA (ethylenediaminetetraacetic acid) and 0.5% BSA (bovine serum albumin)] at 300× *g* for 5 min at room temperature (RT). Supernatant was discarded and cells were resuspended in 1 mL MACS-buffer. Approximately 0.5 × 10^6^ cells were transferred to each well of a round bottom 96-well plate (Sarstedt, Nümbrecht, Germany) and stained with distinct panels of antibodies for 20 min at 4 °C. For strategy A, cells were stained for CD11b, CD14 and CD16 and, for strategy B, with additional FITC-labelled antibodies that bind to CD3, CD209 and CD19. Strategy C used the following antibody combination against CD11b, CD14, CD16 and HLA-DR. Strategies A, B and C were performed on mostly the same blood samples. Samples from patients with STEMI and CHD were stained according to strategy C only. Additional staining for CCR2 and CX3CR1 was performed for healthy controls and patients with STEMI after staining and gating according to strategy C to determine proper identification of monocyte subsets. Of note, strategy C was used to identify monocyte subsets in all functional experiments (phagocytosis of lipid nanoparticles, uptake of 2-NBD-glucose (2-(7-Nitro-2,1,3-benzoxadiazol-4-yl)-D-glucosamine) and intracellular levels of CD74 and HLA-DM). All antibodies were purchased from BioLegend (San Diego, CA, USA) and diluted in MACS-buffer. Please refer to Appendix A for further details (e.g., clone number, fluorochromes) on antibodies used in this study. After incubation with antibodies, cells were washed twice with 500 µL MACS-buffer and stained with 4′,6-diamidino-2-phenylindole (DAPI, 1 µg/mL) in 250 µL MACS-buffer to label dead (DAPI^+^) cells. Finally, cells were analysed by flow cytometry using a BD FACS verse^TM^.

#### 2.3.2. Fixation and Permeabilisation for Intracellular Staining

After sample collection and erythrocyte lysis, cells were first stained for CD11b, CD14, CD16 and HLA-DR, as described in Section 2.3.1. After washing with MACS-buffer, approximately 2 × 10^6^ cells were resuspended in 250 µL fixation buffer (BioLegend, San Diego, CA, USA) for 25 min at RT. Fixed cells were washed once in 1 mL MACS-buffer and twice in 250 µL of an intracellular staining permeabilisation wash buffer (BioLegend, San Diego, CA, USA). Intracellular antibody staining for HLA-DM and CD74 was performed in 100 µL of intracellular staining permeabilisation wash buffer for 20 min at 4 °C (a list of antibodies is shown in Appendix A). The samples were washed again in 500 µL of MACS-buffer, resuspended in 250 µL MACS-buffer and transferred into tubes for flow cytometry.

#### 2.3.3. Cellular Uptake of Perfluorocarbon Nanoemulsions (PFCs)

Approximately 1 × 10^6^ human peripheral blood cells were resuspended in 1 mL of RPMI 1640 medium (PAN-Biotech, Aidenbach, Germany) containing 10% FBS (PAN-Biotech, Aidenbach, Germany), 1 mM sodium pyruvate (Sigma-Aldrich, St. Louis, MO, USA), 2.5 g/L glucose (Carl Roth, Karlsruhe, Germany), 100 U/mL penicillin and 100 g/mL streptomycin (PAN-Biotech, Aidenbach, Germany). An amount of 2 mL of ^A488^PFCs was added to 1 mL of cell suspension and incubated with agitation for 20 min at 37 °C. Afterwards, the cells were centrifuged and washed twice with 1 mL of MACS-buffer before performing antibody staining according to strategy C (CD11b-APC, CD14-PE.Cy7, CD16-APC.Cy7, HLA-DR-PerCP.Cy5.5 and DAPI), as described in Section 2.3.1. To analyse the PFC uptake, the mean fluorescence intensity (MFI) of cells incubated with ^A488^PFCs was measured by flow cytometry and the fluorescence values from untreated samples (background) was subtracted.

#### 2.3.4. Cellular Uptake of 2-NBD-Glucose

Glucose uptake by blood immune cells was analysed with the glucose analogue 2-(7-Nitro-2,1,3-benzoxadiazol-4-yl)-D-glucosamine (2-NBDG, ThermoFisher, Waltham, MA, USA). After sample collection as described in Section 2.3.1, whole blood was incubated with 25 µg/mL 2-NBDG protected from light for 25 min at 37 °C on a shaker. Afterwards, erythrocytes were lysed and cells were then stained for CD11b, CD14, CD16 and HLA-DR and finally analysed by flow cytometry. The MFI was measured, and the fluorescence background from untreated samples was subtracted.

#### 2.3.5. Analysis of Flow Cytometric Data

Flow cytometry was performed using a BD FACS Verse^TM^ and the corresponding BD FACSuite^TM^ software version V1.0.6 (Becton Dickinson, Heidelberg, Germany). Analysis of the flow cytometric data and preparation of the dot plots and histograms were performed using FlowJo V10.8.1 (FlowJo, Ashland, OR, USA) or Kaluza Analysis 2.2.1 (Beckman Coulter, Indianapolis, IN, USA). The gate boundaries were set using fluorescence-minus-one (FMO) controls for each marker of interest.

### 2.4. Statistical Analysis of the Data

Statistical analysis was conducted with GraphPad Prism 10 software (GraphPad Software, Boston, MA, USA). All results were stated as mean values with the according standard deviation. A Shapiro–Wilk test was performed to test for the Gauss distribution of the data points.

To compare two groups, an unpaired two-tailed *t*-test was used if the data showed a normal distribution. If the standard deviation was different between the groups, a *t*-test with Welch correction was conducted. When the data points did not fit to a Gauss distribution, a two-tailed Mann–Whitney U test was used.

For statistical analysis of three groups whose data points were normally distributed, a one-way ANOVA with Tukey’s post hoc test was applied. A Levene’s test was used to assess the equality of variances. When the Levene’s test revealed that the variance differed statistically, an ANOVA with Welch correction was performed.

In case the values did not show normal distribution, the data were analysed with a Kruskal–Wallis test, followed by a Dunn’s post hoc analysis.

Significance was defined as *p*-values ≤ 0.05: * *p* < 0.05, ** *p* < 0.01, *** *p* < 0.001, **** *p* < 0.0001.

## 3. Results

### 3.1. Identification of Monocytes and Monocyte Subsets

To precisely identify monocytes and their subsets in blood samples derived from healthy patients and patients with diseases, we first compared modified versions of three different antibody staining and gating strategies (A, B and C), that were previously used for the identification of monocyte subsets [27,28,29,30]. Blood derived from healthy volunteers was subjected to hypotonic erythrocyte lysis, and subsequently stained with different antibody panels depending on the respective strategy (see below). (The full gating protocol for strategies A-C is provided in Appendix A.)

For strategy A (Figure 1A), cells were stained with antibodies against CD11b, CD14 and CD16. Cells that comprised lymphocytes and monocytes were selected based on their size and granularity (forward/side scatter (FSC/SSC)) properties (Figure 1A, top). Next, we identified CD11b^+^ cells from the first gate and then determined the cell surface expression pattern of CD14 and CD16 in CD11b^+^ cells. Using this strategy, we found that classical monocytes (CD14^++^, CD16^−^) represent 68,9% (±17.9%) of all monocytes, while intermediate monocytes (CD14^+^, CD16^+^) add up to 3.9% (±2.7%) and non-classical monocytes (CD14^+^, CD16^++^) make up 3% (±2.2%). However, in addition to the classical CD14/16 expression pattern of the monocytes, we also observed a relatively prominent CD16^++^ and CD14^−^ contaminating cell population (12.9% ± 8.4%).

Strategy B (Figure 1B), aimed to identify monocytes through the exclusion of other cell types. Here, T cells, B cells and dendritic cells were separated from the CD11b^+^ population by staining the cells with FITC-conjugated antibodies against CD3, CD19 and CD209 (dump gating). We did not include an antibody staining for CD56, which is often used to identify NK cells [31], because approximately 5–10% of the monocytes also expressed CD56 [32] (Appendix A). FITC-negative but CD11b^+^ cells were then used to further distinguish monocyte subsets by CD14/CD16 expression (Figure 1B, third row). The application of strategy B showed that classical monocytes represented approximately 59.1% (±11.6%) and intermediate monocytes and non-classical monocytes accounted for 2.6% (±1.9%) and 3.4% (±2.6%) of all monocytes, respectively. Similar to strategy A, a high ratio of CD16^++^ and CD14^−^ cells (18.7% ± 3.9%) were found close to the non-classical monocytes.

We speculated that the CD16^++^ and CD14^-^ contaminating cell population could be NK cells. It has been described that NK cells can express CD11b but lack CD14 on the cell surface [33]. Analysis of the CD56 cell surface level on the CD16^++^/CD14^−^ population revealed high levels of CD56 (Appendix A). Since NK cells do not express MHC class II molecules [28], we adopted a staining and gating strategy (strategy C) that included the labelling of HLA-DR to exclude NK cells and B cells [28]. We expanded this strategy by adding CD11b to the antibody panel to more precisely eliminate HLA-DR expressing cells, but CD11b-negative B cells. After the gating of cells based on their FCS/SSC properties and CD11b expression, CD14/CD16 double negative cells were excluded (Figure 1C, third row). Next, cells that expressed either CD14 or CD16 or both were plotted against CD16 and HLA-DR to eliminate NK cells that were CD16^high^ but lacked MHC class II. Finally, the expression of CD14/16 was used to identify classical, intermediate and non-classical monocytes (Figure 1C, fifth row). By gating the monocytes as depicted in Figure 1C, about 87.3% (±5.9%) were classical monocytes, whereas intermediate and non-classical monocytes added up to 6.2% (±2.8%) and 4.1% (±2.3%), respectively.

Importantly, the application of strategy C nearly completely abolished the contaminating (CD16^++^, CD14^−^) population (0.4% vs. 15–20% for strategy A or B) compared with strategy A and B (Figure 1D, lower right). Of note, the exclusion of the contaminating CD16^++^/CD14^−^ population led to generally higher ratios of classical monocytes and, to a lower extent, also intermediate monocytes compared with strategy A and B (Figure 1D).

Furthermore, the back gating of monocytes revealed that, in particular, non-classical monocytes had a lower FSC-A value than classical and intermediate monocytes and therefore were more prone to overlap with the lymphocytes in the FSC/SSC plot (Appendix A). Therefore, we decided to include monocytes and lymphocytes in the first SSC/FSC gate to minimize the loss of monocytes during the gating process (Figure 1A–C, first line).

Taken together, strategy C displayed the most reliable way to identify monocytes and their subtypes without any significant contamination of T cells, B cells, dendritic cells or NK cells.

### 3.2. Analysis of Monocyte Subpopulations in Patients with Coronary Heart Disease (CHD) and with ST-Elevation Myocardial Infarction (STEMI)

The precise identification of monocyte subsets is essential for morphological and functional analysis of monocytes and their subsets, especially in states of disease. Cell composition and the expression of surface markers may vary in patients’ samples, making the proper assessment of monocyte subpopulations challenging. To investigate whether strategy C is suitable to precisely identify monocytes and their subtypes under inflammatory conditions, we analysed blood samples from patients one day after acute ST-elevation myocardial infarction (STEMI) and from patients with stable coronary heart disease (CHD).

Blood samples from patients with CHD and STEMI showed a different cellular pattern in the FSC/SSC plot, with a higher number of neutrophil granulocytes (large population with high SSC) compared with healthy volunteers (Figure 2A, left row). Furthermore, the subsequent gates also showed altered expression levels of CD11b, CD14, CD16 and HLA-DR, resulting in shifts of the cell populations (Figure 2A–C, second to fourth columns). Although the expression levels of CD11b/HLA-DR varied in patients with CHD and STEMI, staining and gating according to strategy C could precisely identify monocytes and their subtypes in blood samples from patients with STEMI and CHD (Figure 2B, fifth column). Of note, the gates had to be slightly adjusted because of the altered expression level compositions of the cells in the patients with STEMI and CHD. The quantitative analysis of monocyte subtypes of healthy, STEMI and CHD samples showed a small increase in intermediate monocytes in patients with STEMI (6.9 ± 4%) and CHD (5.6 ± 2.7%) compared with young healthy controls (4.9 ± 1.6%) (Figure 2D). Additionally, statistically significant differences between the groups were found for non-classical monocytes. Here, the highest amount of non-classical monocytes was observed in patients with CHD (6.7 ± 3.2%), whereas only 2.5% (±1.1%) of the monocytes were non-classical in the blood of patients with STEMI and 4.8% (±2.1%) in young healthy donors (Figure 2D, right). (The complete gating protocol for cells obtained from patients with STEMI and CHD is provided in Appendix A).

During clinical routine work, blood samples may not be properly stored or are frequently transferred to the researcher after a prolonged time. To mimic this scenario, we stored the blood of the healthy donors for one hour at room temperature and then analysed the cells by flow cytometry. We observed that the FSC/SSC properties and the appearance in the CD11b/SSC plot was strongly altered (Appendix A). However, staining and gating according to strategy C revealed that it was still possible to reliably identify monocyte subpopulations.

### 3.3. Expression of CCR2 and CX3CR1 in Monocyte Subsets

It is known that the expression of the chemokine receptors CCR2 and CX3CR1 differ among the monocyte subsets [34]. For that reason, we aimed to determine whether the CD14/CD16 gating properly identified monocyte subsets according to their chemokine receptor repertoire. To this end, cells from healthy volunteers and patients with STEMI were isolated and stained as described above (strategy C), and additionally incubated with antibodies that recognize CCR2 and CX3CR1.

The expression levels of CCR2 were higher in all monocyte subsets of patients with STEMI compared with healthy volunteers, with an increase from 12,434 (±6363) to 26,236 (±15,298) in classical monocytes, 4959 (±3228) to 17,397 (±11,712) in intermediate monocytes and 769.2 (±140.6) to 1660 (±1033) in non-classical monocytes. (Figure 3A). The highest expression levels of CX3CR1 were observed in healthy controls compared with patients with STEMI for classical (healthy = 7047 ± 1527; STEMI = 4907 ± 1604), intermediate (healthy = 20,316 ± 3810; STEMI = 11,282 ± 2515) and non-classical (healthy = 25,074 ± 5755; STEMI = 15,501 ± 4577) monocytes (Figure 3B).

The quantitative analysis of CCR2 expression in healthy volunteers and patients with STEMI further revealed the highest fluorescence intensities in classical monocytes (healthy = 12,434 ± 6363; STEMI = 26,236 ± 15,298), followed by intermediate (healthy = 4959 ± 3228; STEMI = 17,397 ± 11,712) and non-classical monocytes (healthy = 769.2 ± 140.6; STEMI = 1660 ± 1033) (Figure 3A). The CCR2 expression differed between classical monocytes and non-classical monocytes and between intermediate monocytes and non-classical monocytes in healthy volunteers and patients with STEMI.

The expression levels of CX3CR1 were lowest in classical monocytes (healthy = 7047 ± 1527; STEMI = 4907 ± 1604), and higher in intermediate (healthy = 20,316 ± 3810; STEMI = 11,282 ± 2515) and non-classical monocytes (healthy = 25,074 ± 5755; STEMI = 15,501 ± 1577) in blood samples from healthy volunteers and patients with STEMI. For samples obtained from healthy volunteers and patients with STEMI, the CX3CR1 expression levels differed between classical, intermediate and non-classical monocytes.

Taken together, the CCR2/CX3CR1 surface levels supported our initial finding that strategy C is suitable to clearly identify monocyte subsets in both healthy and disease states.

### 3.4. Identification of Monocyte Subtypes after Fixation and Permeabilisation

Monocyte subtypes also differ in functional aspects, which are also represented by intracellular alterations such as cytokine secretion, signal transduction or antigen processing and presentation [9,10]. Assessment of these processes in monocyte subtypes by flow cytometry requires fixation and permeabilisation of the cells, which can strongly alter the FSC/SSC properties and the detection levels of marker molecules.

To investigate intracellular processes in monocyte subpopulations, we utilised the staining and gating protocol of strategy C and then subjected the cells to fixation and permeabilisation (see method Section 2.3.2 for details). The FSC/SSC-based morphology of fixed blood cells was quite similar to untreated cells but showed a decreased SSC after permeabilisation (Figure 4A–C, left panel). This resulted in a closer proximity of lymphocytes, particularly neutrophil granulocytes, to monocytes in the FSC/SSC panel. However, gating for CD11b, CD14 and CD16 and HLA-DR and then plotting of CD14 against CD16 allowed for a clear separation of monocyte subsets, resulting in a similar distribution of monocyte subsets in permeabilised blood samples compared with untreated samples (Figure 4A–C, right panel). The number of classical monocytes varied from 90% (±3.7%) to 93.2% (±2.3%) and 90.4% (±4%) between untreated, fixed and permeabilised samples. For intermediate monocytes, mean percentages of 4.8% (±1.7%) in untreated, 3.7% (±1.3%) in fixed and 4.5% (±1.4%) in permeabilised samples were detected (Figure 4D). The relative amount of non-classical monocytes was 4.6% (±2.1%) in untreated cells, 2.6% (±1%) in fixed and 4.4% (±2.6%) in permeabilised samples. Finally, the identification of monocyte subsets using strategy C was further verified by the staining of fixed and permeabilised blood samples of healthy volunteers and patients with STEMI against CCR2 and CX3CR1 (Appendix A).

In summary, this fixation and permeabilisation protocol enabled the precise identification of the monocyte subsets and identical composition of classical, intermediate and non-classical monocytes compared with unfixed cells when using staining strategy C.

### 3.5. Assessment of Phagocytosis and Glucose Uptake in Monocyte Subsets

Next, we wanted to assess the functional aspects of monocyte subtypes that were identified by strategy C. To this end, we utilised flow cytometry to analyse phagocytosis/endocytosis of fluorescent lipid nanoparticles (^A488^PFCs) and the uptake of the glucose analogue 2-NBDG in monocyte subsets from healthy controls and patients with stable CHD or one day after STEMI.

First, we determined the cellular uptake of fluorescently labelled ~180 nm sized lipid nanoparticles (perfluorocarbon nanoemulsions = ^A488^PFCs) prepared in house (see methods and Supplemental Appendix A). Human whole blood was incubated with ^A488^PFCs at 37 °C. The flow cytometric analysis revealed a heterogeneous pattern of PFC uptake (Figure 5A). Determination of the fluorescence signal of cell-associated ^A488^PFCs revealed an MFI of 135 (±30.6) in healthy volunteers, 189 (±50) in patients with CHD and 45 (±19) in patients with STEMI. A similar pattern in the ^A488^PFC uptake was observed in intermediate monocytes, with the highest uptake in patients with CHD (155 ± 55.3), followed by healthy volunteers (111 ± 30.7) and patients with STEMI (47 ± 21.1). Interestingly, non-classical monocytes from healthy volunteers showed nearly no uptake, whereas there was low labelling of the cells derived from patients with CHD and STEMI (CHD = 26 ± 23; STEMI = 6.4 ± 7.4).

It has been proposed that glucose uptake by monocytes is an important feature in inflammatory disease, as the activation of monocytes promotes a switch from an oxidative metabolism to a glycolytic metabolism [35]. Glucose uptake can be analysed with the fluorescent glucose analogue 2-NBDG [35]. Human whole blood was incubated with 2-NBDG and stained according to gating strategy C to identify monocyte subsets. Measuring the MFI of the 2-NBDG signal in classical monocytes, the highest fluorescence signal was detected in STEMI samples (301 ± 194.9) compared with blood from healthy volunteers (162.7 ± 69.1) and patients with CHD (140.3 ± 31.2) (Figure 5B). A similar pattern was observed in intermediate monocytes, where STEMI samples showed the highest MFI at 318.4 (± 208.1) compared with 139.1 (±59.0) in cells from healthy controls and 137.4 (±28.46) in patients with CHD. In all groups, the overall uptake of 2-NBDG was lower in non-classical compared with classical and intermediate monocytes. Nonetheless, the non-classical monocytes from patients with STEMI again displayed the highest uptake of 2-NBDG (MFI = 208.1 ± 146.7) compared with healthy volunteers and CHD samples (healthy control = 101.0 ± 40.8; CHD = 93.9 ± 26.7).

### 3.6. Intracellular Analysis of Components of the MHC-Class II Pathway

Another important function of monocytes is the processing and presentation of antigenic peptides by the MHC class II pathway, which is crucial for the activation of antigen-specific CD4^+^ T cells. To gain insights into the class II pathway of antigen presentation, the expressions of two main molecules, CD74 and HLA-DM, were analysed on the cell surface and intracellularly in monocyte subsets from healthy volunteers. CD74 is involved in the assembly and transport of MHC class II molecules to endosomal compartments and HLA-DM acts as a chaperon to load high-affinity peptides onto MHC class II molecules [36,37]. HLA-DM and CD74 were strongly expressed intracellularly and to a lower degree on the cell surface (Figure 6A,B). Classical and intermediate monocytes showed a similar intracellular CD74 expression with an MFI of 15,074 (±7257) and 15,056 (±8857), respectively, whereas non-classical monocytes seemed to express lower levels of CD74 with an MFI of 8323 (±3457) (Figure 6C, left). In comparison with classical (12,245 ± 3913) and non-classical monocytes (12,596 ± 5142), the highest HLA-DM expression was observed in intermediate monocytes with an MFI of 33,550 (±8817) (Figure 6C, right).

## 4. Discussion

Monocytes come from a heterogeneous cell population that plays an important role in the protection against infectious diseases, and monocytes are also involved in tissue repair and wound healing after sterile injury. However, the precise role of individual monocyte subtypes for the development and progression of many inflammatory diseases is still incompletely understood. Unravelling their role requires the unequivocal identification of monocytes and their subtypes and the analysis of their phenotypical and functional properties. Here, we suggest a robust flow cytometric method for the precise identification and functional analysis of monocyte subtypes that was validated in healthy subjects, but also in patients with stable CHD and STEMI.

### 4.1. Flow Cytometry of Monocyte Subtypes

In the present study, we compared three different staining and gating strategies (strategies A–C) for the identification of human monocytes and their subtypes. To this end, we utilised antibodies against CD11b, HLA-DR and CD14 and CD16 followed by several gating steps (strategy C) to identify classical, intermediate and non-classical monocyte subtypes in healthy subjects, and also in patients with CHD and STEMI. Monocytes and their subtypes were identified according to their FSC/SSC properties (morphology), the expression of CD11b, HLA-DR and the differential expression pattern of CD14/CD16 [22,28,38]. Staining for CD11b and HLA-DR enabled us to particularly exclude B cells and NK cells. B cells are CD11b negative, but express high levels of MHC class II molecules, whereas parts of NK cells can express low levels of CD11b but are negative for HLA-DR. Of note, the CD11b expression varies among the monocyte subtypes. Specifically, non-classical monocytes express lower levels of CD11b, which makes it possible that some monocytes with a low expression of CD11b could be lost by gating on CD11b^+^ cells. Another aspect is that the expression of CD11b is dynamic and can either be upregulated under inflammatory conditions [39] or downregulated, for example, by n-butyrate [40].

NK cells can severely contaminate the low abundant population of the non-classical monocytes (~1% of all leukocytes) because they can make up about 5% of the blood immune cells, and express high levels of CD16 and lack CD14 [28,41]. Staining and gating strategies A and B revealed strong contaminations with CD16^++^/CD14^−^ NK cells in the CD14/CD16 plot, which was completely abolished by the application of strategy C (Figure 1). One reason for the high number of contaminating NK cells in strategies A and B was that we very broadly selected a population that contained lymphocytes and monocytes in the first FSC/SSC gate. Several studies suggest to use a very narrow gate in the FSC/SSC plot to remove all non-monocytes [20,21,42]. However, this can result in the loss of monocytes, in particular, monocytes with low FSC/SSC properties. Back gating revealed that a fraction of the non-classical monocytes showed low FSC-A values, which were therefore more closely located to the lymphocyte population, which can result in a partial overlap of these cells (Appendix A). As already mentioned above, NK cells can contaminate the monocyte population and the NK marker CD56 is often used for the exclusion of these cells. However, about 5–10% of the monocytes of healthy persons expressed CD56 (Appendix A); although, the expression levels were lower than for conventional NK cells (Appendix A). The population of CD56^+^ monocytes contained higher levels of proinflammatory molecules like TNFα, IL-10 and IL-23 upon restimulation with LPS [32]. Furthermore, high numbers of CD56^+^ monocytes have been found in patients with rheumatoid arthritis [32], Crohn’s disease [43] or cancer [44]. Because of this, we did not use CD56 as an NK cell exclusion marker, but, in other situations, the exclusion of CD56^+^ cells could be considered.

Since we set a very generous FSC/SSC gate to minimise the loss of monocytes, we used HLA-DR molecules as additional markers for the identification of monocytes. HLA-DR molecules were not expressed by NK cells or neutrophils and the combination of CD11b and HLA-DR, followed by sequential gating, eliminated the contamination of CD16^++^/CD14^−^ and CD16^−^/CD14^−^ cells from the CD14/16 plot (Figure 1D). This strategy was not only valid in blood samples from healthy subjects but also in blood from patients with CHD or STEMI (Figure 2). It is of crucial importance to inspect the staining and gating of immune cell subsets also under diseased conditions, as the expression of cell surface markers and the proportion of the immune subtypes can be strongly altered under inflammatory conditions. For example, it is known that acute myocardial infarction (MI) results in a strong increase in circulating monocytes and neutrophils and altered expression levels of cell surface markers [45]. Here, we observed an increased expression of CD11b in monocytes from patients with CHD and STEMI, which indicated the activation of the cells [46] but did not disturb the identification of monocyte subtypes (Figure 2). Other studies have suggested to use CD86 as a third marker for identification of monocytes [18,19]. But, it has been reported that the expression of CD86 is variable under inflammatory conditions [47]. Jurado et al. developed a 10-colour panel which included CD45, CD33, CD14, CD16, CD64, CD82, CD300, CD2, CD66c and CD56 to analyse monocyte subtypes from patients with myelomonocytic leukaemia [48]. Another study developed a protocol (OMIP 083) that included 21 markers with 18 colours for a highly sophisticated in-depth analysis of monocytes and their subtypes [22]. These approaches allow for a deeper analysis and the identification of further monocyte subpopulations but are not suitable for flow cytometers, with less capabilities concerning their lasers and channels. In addition, when using conventional machines, it leaves very little space for further phenotypical or functional analyses, such as cytokine expression, cell signalling or assessment of endocytosis/phagocytosis.

### 4.2. Phenotype and Function of Monocyte Subtypes in Health and Disease

Multiple studies have shown that the composition, the phenotype and the functionality of monocyte subtypes can be altered under inflammatory conditions and that these alterations can affect the pathophysiology of certain diseases. In this study, we demonstrated that gating strategy C was suitable for identifying monocytes and their subsets in blood samples from patients with stable CHD or one day after STEMI despite alterations in the cell composition (e.g., high numbers of neutrophils). In patients with STEMI, a significant decrease in numbers of non-classical monocytes and a slight but non-significant increase in intermediate monocytes were detected compared with healthy and CHD samples (Figure 2). These results are partly reflected by studies from Zhou et al. [49] and Tapp et al. [50], who found increased intermediate monocytes one day after STEMI and no changes in non-classical monocytes compared with a healthy control. Wrigley et al. [51] discovered elevated classical monocytes and intermediate monocytes in patients with acute and chronic heart failure. Increased numbers of classical monocytes have also been found in patients with rheumatoid arthritis [52], or in patients undergoing haemodialysis [53].

ST-elevation myocardial infarction is associated with the death of a large number of cardiomyocytes and the release of damage-associated molecular patterns (DAMPs) [54]. These DAMPs initiate a strong proinflammatory reaction that also leads to the activation of monocytes [55]. In agreement with this, we found an increased uptake of 2-NBDG in all monocyte subsets in patients with STEMI compared with healthy volunteers and patients with CHD (Figure 5B). A study by Palmer et al. [35] revealed an equal uptake of 2-NBDG in classical, intermediate and non-classical monocyte subsets. However, this study also found an increased 2-NBDG uptake in all subsets in patients with HIV compared with healthy controls [35]. These findings are in line with the concept that the activation of monocytes results in an increased uptake of glucose and a switch from an oxidative to a glycolytic metabolism [56]. Although it has been reported that monocytes in patients with CHD display an activated phenotype [57], the uptake of 2-NBDG was similar to healthy human volunteers. However, Dietl et al. found only a slight increase in glucose uptake in LPS-stimulated monocytes but a strongly increased generation of lactic acid [56]. Therefore, it could be possible that patients with CHD have activated monocytes that do not take up larger amount of glucose but switch towards a glycolytic metabolism.

The activation of monocytes is often associated with increased endocytic/phagocytic properties. Here, we utilised fluorescent lipid nanoparticles (^A488^PFCs) manufactured in house with a hydrodynamic diameter of ~180 nm to investigate endocytosis/phagocytosis in monocyte subtypes (Figure 5A). Interestingly, all monocyte subtypes from patients with CHD showed a higher PFC uptake than healthy volunteers, whereas monocytes from patients with STEMI displayed the lowest endocytosis of PFCs. The PFC uptake of classical monocytes and intermediate monocytes was comparable, but non-classical monocytes showed low labelling with PFCs. In fact, non-classical monocytes from healthy persons did not take up any PFCs, whereas non-classical monocytes from patients with CHD and STEMI showed a low amount of uptake. The findings are largely in agreement with our recent study where we investigated the uptake of fluorescently labelled PFOB-PFCs by monocyte subtypes from patients with STEMI and CHD [30]. The strongest PFC uptake was found in intermediate monocytes, followed by classical monocytes. Non-classical monocytes internalised only very low levels of PFCs. Nienhaus et al. did not observe any difference in PFC uptake between monocytes from healthy volunteers and patients with STEMI and CHD [30]. In contrast, the present study found that monocytes from patients with CHD had a higher uptake of PFCs compared with healthy human volunteers and patients with STEMI. One explanation could be that the hydrodynamic diameter of the PFCs was different. Here, we used PFCs with a diameter of ~180 nm and the former study was conducted with PFCs that had a size of 210 nm. Of note, the uptake of these PFOB-PFCs required active phagocytosis, as shown by the treatment of the cells with cytochalasin D [30]. Another difference was that we used monocytes obtained on the first day post MI, whereas the previous study was conducted with monocytes that were derived on day three post MI.

Another important function of monocytes is the processing of antigenic peptides via the MHC class II pathway and the presentation of small antigenic peptides to CD4^+^ T cells with a matching T cell receptor [58,59]. To gain an insight into the MHC class II pathway, we analysed the intracellular levels of two main players in the antigen processing pathway—CD74 and HLA-DM. To this end, we established a fixation and permeabilisation protocol that allows for the unequivocal identification of monocyte subtypes. Cells were first stained with antibodies against CD11b, HLA-DR, CD14 and CD16 and afterwards fixed, permeabilised and stained for intracellular markers. The permeabilisation process with solvents such as saponin, Triton X-100 or Tween-20 results in the removal of cholesterol and lipids from the cell membrane, thus allowing antibodies to reach targets inside the cell [60]. The disruption of the cell membrane integrity can lead to the loss of proteins on the cell surface and an altered cell morphology. We observed changes in the FSC/SSC properties after fixation and permeabilisation, but the staining and gating of strategy C revealed only minor differences in the composition of the subtypes in fixed and permeabilised cells (Figure 4). Importantly, the precise distinction of monocyte subtypes was also verified in samples from patients with STEMI and by analysis of CCR2 and CX3CR1 expression (Appendix A). Our results showed that CD74 and HLA-DM were strongly expressed intracellularly and to a lesser extent on the cell surface (Figure 6). The intracellular expression of CD74 was higher in classical and intermediate monocytes compared with non-classical monocytes. The highest levels of HLA-DM were observed in intermediate monocytes compared with classical and non-classical monocytes. These findings are in line with a study from Wong et al. [34], who utilised gene expression profiling to show that intermediate monocytes express higher levels of markers involved in antigen-presenting cell and T cell interactions. In this and other studies, it has also been found that intermediate monocytes demonstrate a high degree of MHC class II processing and antigen presentation [19,34].

### 4.3. Drawbacks of the CD14/CD16 Classification of Monocyte Subtypes

In the present study, we utilised the widely accepted CD14/CD16 classification system [38] to discriminate between classical, intermediate and non-classical monocytes. However, the CD14/CD16 system also has limitations, because the expression of these molecules can vary under inflammatory conditions. For example, the stimulation of isolated and cultured monocytes with LPS results in a rapid downregulation of CD16, making it impossible to discriminate classical, intermediate and non-classical monocytes. Because of this, Ong et al. utilised CyTOF (cytometry by time of flight) of 34 markers and found an alternative panel composed of CD64, CD86, CD33 and CCR2 to investigate monocyte subtypes [47]. Using this antibody panel, the authors identified three different monocyte subsets, and they also found that, in patients suffering from Dengue fever, classical monocytes upregulated CD16, which can result in the misinterpretation of these cells as intermediate or non-classical monocytes. In our study, we did not find major alterations in CD14/CD16 expression levels in patients with CHD and STEMI that could render the differentiation of the three subtypes difficult. To inspect our gating strategy, we analysed the expression of the chemokine receptors CCR2 and CX3CR1 and found high levels of CCR2 in classical monocytes and high levels of CX3CR1 in non-classical monocytes (Figure 3), which is in agreement with previous studies [34]. Regardless of the strategy for identifying monocyte subtypes, one should consider that intermediate and non-classical blood monocytes are derived from classical monocytes that differentiate into intermediate and finally into non-classical monocytes [7]. Therefore, blood monocytes represent a continuum rather than clearly separated subsets. Nevertheless, despite its limitations, CD14/CD16 is widely accepted and has been used to determine differences in monocyte subtypes in multiple diseases. In our opinion, it represents a suitable reference system, in particular, in settings where superior performance flow cytometric equipment (e.g., spectral flow or CyTOF) is not available.

### 4.4. Limitations of this Study

The main aim of this study was to develop a robust and simple flow cytometric procedure for the analysis of human monocytes from healthy persons and persons with diseases. To test the principal validity of our approach, we utilised blood samples from young healthy volunteers and patients with CHD and STEMI. It is important to note that our groups were not matched by age or sex. It is known that there are profound immunological differences between young and old and male and female persons. Therefore, we cannot exclude that the differences we observed between the groups could also have been due to other confounding factors.

The second point is that we used HLA-DR as an important marker molecule to exclude NK cells. However, HLA-DR can be severely downregulated in sepsis, and low levels of HLA-DR correlate with mortality or an increased number of infections [61]. In the case of severe downregulation of MHC class II molecules, it has to be evaluated whether the expression level of HLA-DR is still sufficient to discriminate monocytes from NK cells or neutrophils. If not, other markers such as CD86 could be feasible, or alternative approaches, as, for example, suggested by Ong et al. [47], have to be considered.

### 4.5. Conclusions and Outlook

The main aim of this study was to establish a robust and reproducible flow cytometric approach to analyse monocyte subtypes in blood samples from healthy volunteers and patients with diseases. By the application of antibodies against CD11b, HLA-DR, CD14 and CD16 followed by sequential gating steps, we were able to precisely identify classical, intermediate and non-classical monocytes in healthy persons and patients with CHD and STEMI. We found that the composition and the functionality of monocyte subtypes were altered in patients with CHD and STEMI. Moreover, monocyte subtypes differed in the uptake of lipid nanoparticles, the internalisation of 2-NBD-glucose and the intracellular molecules of the MHC class II pathway of antigen presentation. Of note, our flow cytometric approach can be easily extended to more deeply analyse the functionality of monocyte subtypes (e.g., cytokine expression, cell signalling, ROS production) in patients with CHD and STEMI and to gain further insights into the relevance of the monocyte subtypes for the pathophysiology of the diseases.

## Figures and Tables

**Figure 1 biomolecules-14-01251-f001:**
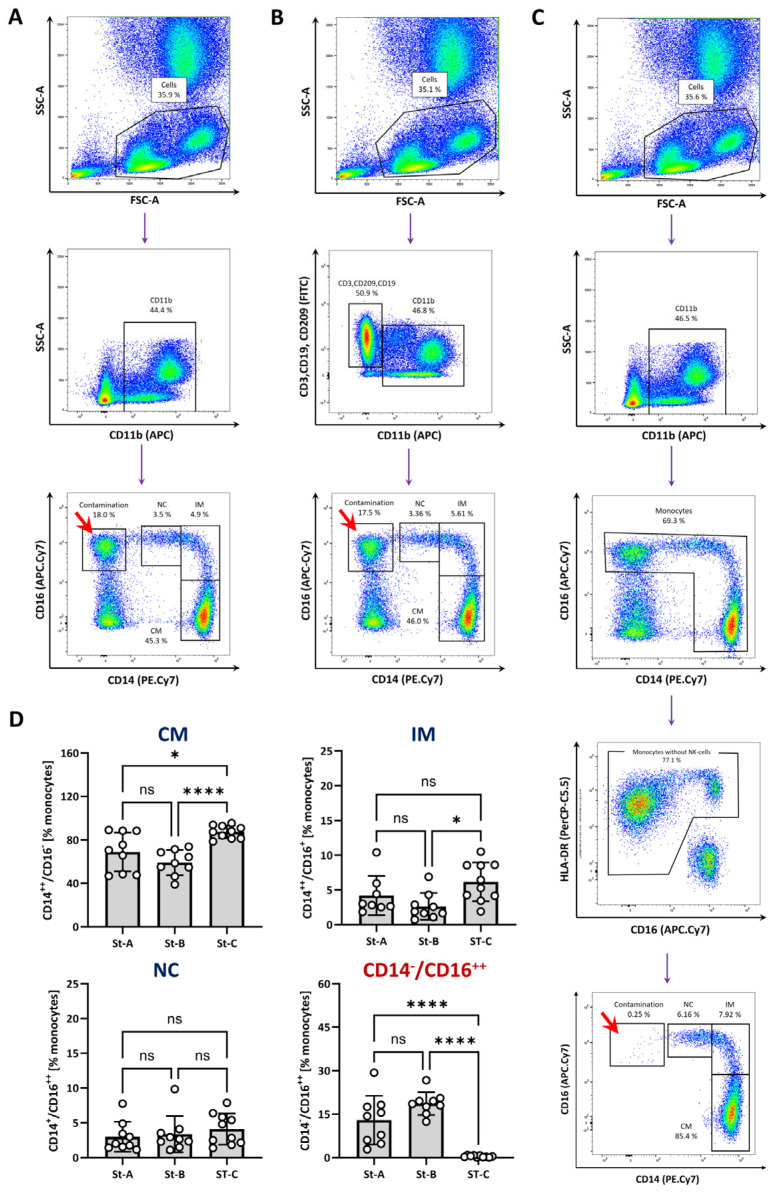
Antibody staining and gating strategies for identification of monocyte subsets. Human whole blood was obtained from healthy volunteers, erythrocytes were lysed and cells were stained with (**A**) CD11b-APC, CD14-PE.Cy7, CD16-APC.Cy7 (strategy A; ST-A); (**B**) CD11b-APC, CD14-PE.Cy7, CD16-APC.Cy7 and CD3-FITC, CD19-FITC, CD209-FITC (strategy B; ST-B); (**C**) CD11b-APC, CD14-PE.Cy7, CD16-APC.Cy7, HLA-DR-PerCP.Cy5.5 (strategy C; ST-C). Blood cells were additionally stained with DAPI (4′,6-Diamidin-2-phenylindol) to exclude dead cells (DAPI^+^) from the analysis (Appendix A). Displayed are pseudo-colour plots that indicate the gating strategy for the identification of monocyte subtypes based on the expression levels of CD14 and CD16. CM—classical monocytes, IM—intermediate monocytes, NC—non-classical monocytes. (**D**) Quantitative analysis of monocyte subset (% of monocytes for each subset and the CD16^+^/CD14^−^ contamination of the monocyte population; marked by red arrows). Mean ± SD are shown; *n* = 9–10. Normality was tested using the Shapiro–Wilk test and variability with Levene’s test. Differences between the groups were analysed by a one-way ANOVA followed by Tukey’s multiple comparisons test. Statistical significance: ns— not significant, * *p* < 0.05, **** *p* < 0.0001. There were 12 healthy volunteers (2 × male, 10 × female; 22–28 years, median: 23 years). The complete gating for identification of monocyte subsets by strategies A–C is shown in Appendix A.

**Figure 2 biomolecules-14-01251-f002:**
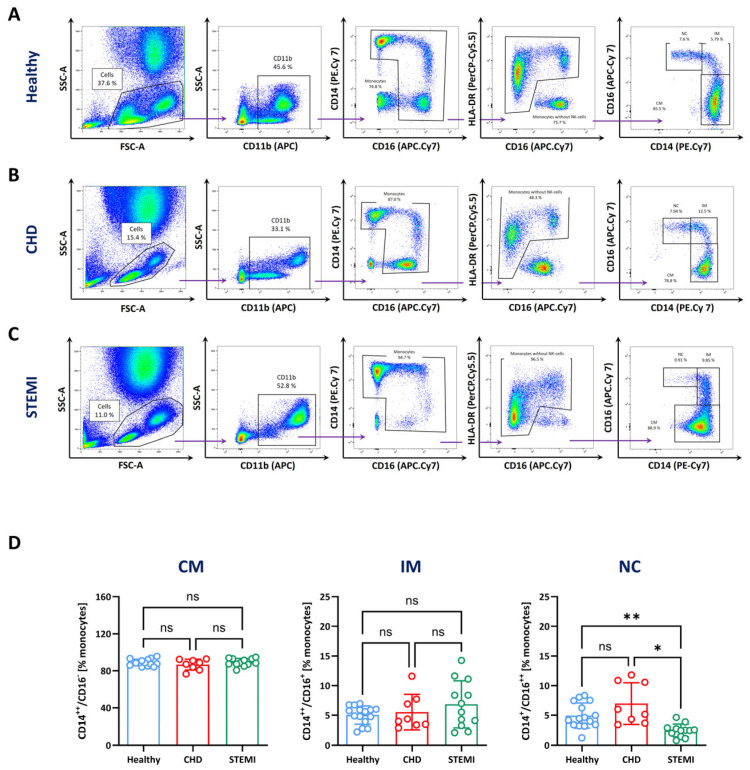
Analysis of monocyte subsets in CHD and STEMI patients. Blood samples derived from healthy volunteers (**A**, blue), patients with stable coronary heart disease (CHD, red) (**B**) and patients with ST-elevation myocardial infarction (STEMI, green) (**C**) were stained for CD11b, CD14, CD16 and HLA-DR to identify monocyte subsets. Displayed are pseudo-colour plots that show the gating strategy (strategy C) and the identification of the monocyte subtypes based on the expression of CD14 and CD16. (**D**) Quantitative analysis of the relative amount of monocyte subsets (CM—classical, IM—intermediate, NC—non-classical) in varying conditions. Data are mean values ± SD of *n* = 10–15. Normality was tested using the Shapiro–Wilk Test and variability with Levene’s test. Differences between the groups were analysed by a one-way ANOVA followed by Tukey’s multiple comparisons test. Statistical significance: ns—not significant, * *p* < 0.05, ** *p* < 0.01. There were 15 healthy controls (3 × male, 12 × female; 22–33 years, median: 24 years), 10 patients with CHD (5 × female, 5 × male; 52–86 years, median: 64 years) and 12 patients with STEMI (1 × female, 11 × male; 40–78 years, median: 60.5 years). (The complete gating strategy for the identification of monocyte subsets is shown in Appendix A).

**Figure 3 biomolecules-14-01251-f003:**
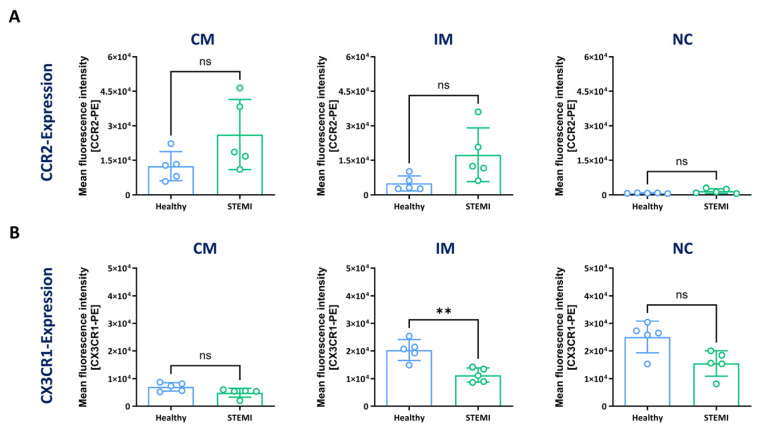
Cell surface expression of CCR2 and CX3CR1 in monocyte subsets. Blood samples were derived from healthy volunteers and from patients one day after ST-elevation myocardial infarction (STEMI). Cells were stained with antibodies against CD11b, CD14, CD16 and HLA-DR (strategy C) to identify monocyte subsets. Cells were then additionally incubated with either anti-CCR2 or anti-CX3CR1 antibodies. Quantitative analysis of CCR2 (**A**) or CX3CR1 (**B**) expression in monocyte subsets of healthy donors (blue) and patients with STEMI (green). Data are mean values ± SD of *n* = 5 individual experiments. Normality was tested using the Shapiro–Wilk test. Differences between the groups were analysed by an unpaired two-tailed *t*-test with or without Welch correction or a Mann–Whitney test. Statistical significance: ns— not significant, ** *p* < 0.01. There were 5 healthy controls (4 × female, 1 × male; 22–27 years, median: 24 years) and 5 patients with STEMI (5 × male; 59–68 years, median: 63 years).

**Figure 4 biomolecules-14-01251-f004:**
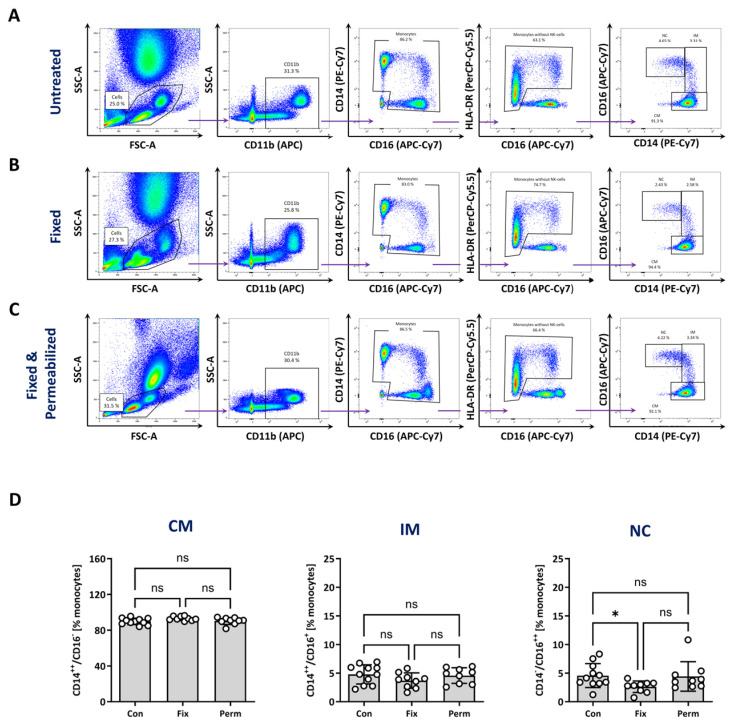
Identification of monocyte subsets after fixation and permeabilisation. Blood was derived from healthy volunteers and cells were stained with antibodies against CD11b, CD14, CD16 and HLA-DR for monocyte identification. Afterwards, cells were left untreated ((**A**), upper panel), fixed at room temperature for 25 min ((**B**), middle panel) or fixed and permeabilised (20 min at room temperature) ((**C**), lower panel). (**D**) Quantitative analysis of the relative amount of monocyte subsets (CM—classical, IM—intermediate, NC—non-classical) after fixation (Fix) and permeabilisation (Perm) of untreated control cells (Con). Normality was tested using a Shapiro–Wilk test and variability with Levene’s test. Differences between the groups were analysed by a one-way ANOVA followed by Tukey’s multiple comparisons test. Data are mean values ± SD of *n* = 9–11 (9 × female, 2 × male; 22–27 years, median: 24 years). Statistical significance: ns—not significant, * *p* < 0.05.

**Figure 5 biomolecules-14-01251-f005:**
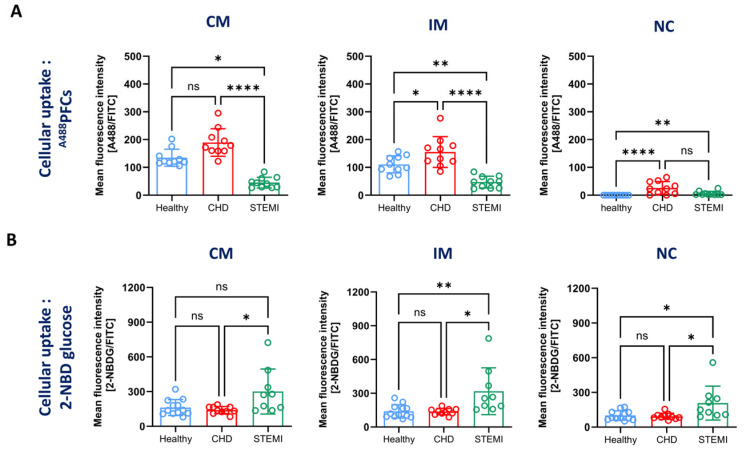
Assessment of phagocytosis and glucose uptake in monocyte subsets. Blood samples were derived from healthy volunteers (blue), patients with stable coronary heart disease (CHD, red) and patients with ST- elevation myocardial infarction (STEMI, green). After lysis of erythrocytes, cells were either incubated with fluorescently labelled perfluorocarbon nanoemulsions (^A488^PFCs) (**A**), or the glucose analogue 2-NBDG [2-(7-Nitro-2,1,3-benzoxadiazol-4-yl)-D-glucosamine] (**B**). Cells were then stained for CD11b, HLA-DR, CD14 and CD16 to determine the uptake of ^A488^PFCs or 2-NBDG in classical (CM), intermediate (IM) and non-classical monocytes (NC) by flow cytometry. Displayed is the quantification of the mean fluorescence intensity of the ^A488^PFCs (**A**) or the 2-NBDG (**B**) signals in CM, IM and NC of healthy volunteers and patients with CHD and STEMI. Data are mean values ± SD of *n* = 9–12. Normality was tested using the Shapiro–Wilk test and variability with Levene’s test. Differences between the groups were analysed by a one-way ANOVA followed by Tukey’s multiple comparisons test. Statistical significance: ns—not significant, * *p* < 0.05, ** *p* < 0.01, **** *p* < 0.0001. (**A**) There were 10 healthy controls (5 × male, 5 × female; 22–45 years, median: 23 years), 10 patients with CHD (5 × male, 5 × female; 52–86 years, median: 64 years) and 10 patients with STEMI (2 × female, 8 × male; 41–80 years, median: 59.5 years). (**B**) There were 12 healthy controls (10 × female, 2 × male; 22–33 years, median: 25 years), 10 patients with CHD (5 × female, 5 × male; 52–86 years, median: 64 years) and 9 patients with STEMI (1 × female, 8 × male; 40–78 years, median: 57 years).

**Figure 6 biomolecules-14-01251-f006:**
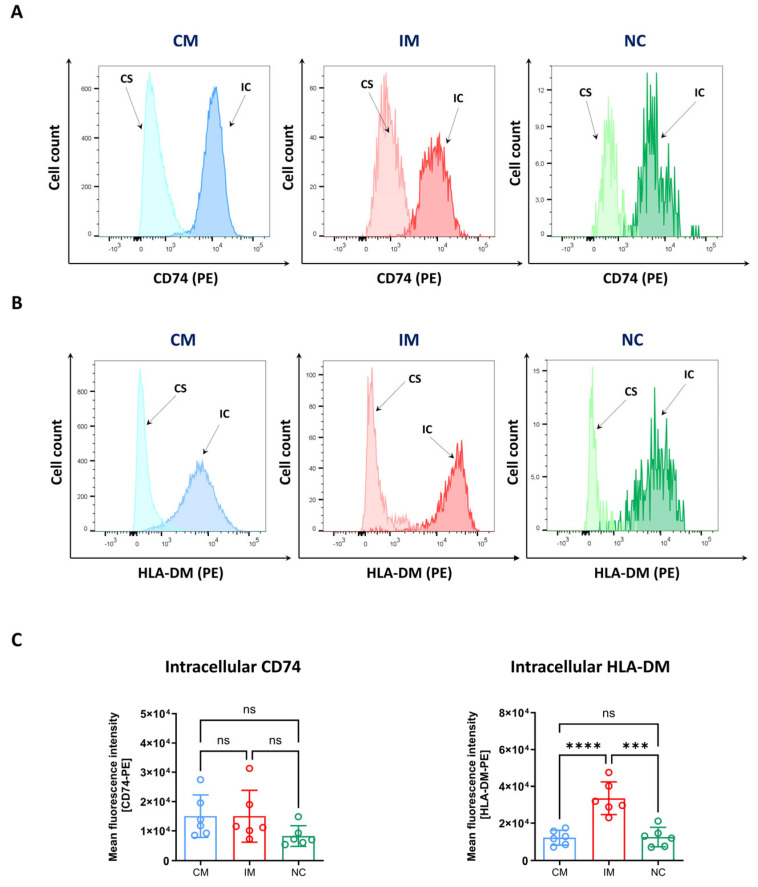
Intracellular staining of HLA-DM and CD74 in monocyte subsets. Blood samples were derived from young healthy volunteers. Cells were first stained with antibodies against CD11b, CD14, CD16 and HLA-DR to identify classical (CM, blue), intermediate (IM, red) and non-classical monocytes (NC, green). Subsequently, the samples were additionally stained for CD74 (**A**) or HLA-DM (**B**) to determine the surface expression (CS). Alternatively, immune cells were fixed and permeabilised and treated with anti-CD74 and HLA-DM antibodies to detect intracellular (IC) expression levels. Light colours represent the expression of CD74 and HLA-DM on the cell surface, darker colours depict intracellular expression. (**C**) Quantitative analysis of the mean fluorescence intensities (MFI) of intracellular CD74 (left) and HLA-DM (right). Data are mean values ± SD, *n* = 5–6 (4 × female, 2 × male; 22–33 years, median: 25 years). Normality was tested using a Shapiro–Wilk test and variability with Levene’s test. Differences between the groups were analysed by a one-way ANOVA followed by Tukey’s multiple comparisons test. Statistical significance: ns—not significant, *** *p* < 0.0001, **** *p* < 0.0001.

## Data Availability

The data presented in this study are available on request from the corresponding author.

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
