# Peer review of "Design of a Robust Flow Cytometric Approach for Phenotypical and Functional Analysis of Human Monocyte Subsets in Health and Disease"

_biomolecules, 2024, doi:10.3390/biom14101251_

Round 1

Reviewer 1 Report

Comments and Suggestions for Authors

In their study, Ahrazoglu and colleagues describe three different staining and gating strategies for the flow cytometry evaluation of the three main monocyte subpopulations. To identify the optimal strategy, they also compare functional assays and the expression of additional phenotypic markers. Finally, they apply the selected best protocol to compare healthy subjects with two groups of patients: one affected by coronary heart disease and the other by ST_elevation myocardial infaction, describing the difference in distribution, functionality, and marker expression in the 3 monocyte subpopulations.

The authors do not describe the source of volunteer samples or whether they were sex- and age-matched with patient groups.

Several references to tables and paragraphs are incorrect and need to be checked and properly corrected. Moreover, the supplementary table antibodies detailing antibodies should include all the antibodies  used in the study.

In figure 1, the axes in the third dot plot of column C should be inverted for consistency within the figure.

I am not sure that the ratio between Atto488 and PFC is correct (100ug per 10g). In section 2.5, it should be specified that antibody staining described in 2.2 corresponds to the Strategy C.

The author tested their analysis panel on sample kept for 1h at RT to mimic laboratory conditions. Out of curiosity, did they test longer time periods? In larger centers, the time between sample collection and delivery to the lab can be even longer.

In nearly every experimental plan, the main result consists in the MFI difference between the 3 patient groups. How do the authors ensure the comparability of data in this type of longitudinal study? How do they minimize variations in laser power from day to day?

In the phagocytosis/endocytosis experiment, it is unclear whether the MFIs reported are from treated samples with or without background subtraction of untreated sample (delta MFI).

Some axis titles contain typos. Maximum values on histogram axes should be the same, where possible, for comparing the same parameter (for example, in Figure 3, the upper panel versus its respective lower panel).

Strategy B lacks a marker to exclude granulocytes, which in several pathologies may have reduces scatter that overlaps with the monocyte scatter region more significantly than NC overlap with the lymphocyte region. Additionally, while CD56 can be expressed on monocytes, as correctly noted by the authors, it is typically found only on pathological monocytes (for example, in CMML) and at a significantly lower antigenic density compared to NK cells. Therefore, in my opinion, CD56 should be considered in an “exclusion panel” setting.

Still regarding strategy B, the positioning of gates as shown in Fig 1 (Column B, second panel) is incorrect: the dump channel is intended to exclude positivity to lineage markers (in this case, CD19, CD3, CD209) co-expressed with the marker used to enrich the population of interest (CD11b in this study).

HLA-DR can be downregulated in monocytes, particularly in cases of immuneparalysis and sepsis, and this important limitation should be discussed in section 4.3.

Comments on the Quality of English Language

Please check for typos throughout the paper, along with the correct chemical nomenclature of the reagents used. Moreover, the punctuation needs to be carefully reviewed.

Reviewer 2 Report

Comments and Suggestions for Authors

Reviewer 3 Report

Comments and Suggestions for Authors

This is a very interesting, well designed and well presented work about the flow cytometric approach for phenotypical and functional analysis of monocyte subsets in health and disease.

It is very clear and easy to be used in cytometry labs all over the world.

A minor typing mistake in line 54, release instead of released. Perhaps the authors should check for typing errors in the whole paper.
